# Ten years of visual field change in people living with diabetes: A prospective longitudinal study

**Karl-Johan Hellgren** [1,2]*, **Boel Bengtsson**[3]

**1** Department of Ophthalmology, Karlstad Hospital, County Council of Värmland, Karlstad, Sweden,
**2** School of Medical Sciences, Faculty of Medicine and Health, Örebro University, Örebro, Sweden,
**3** Department of Clinical Sciences in Malmö, Ophthalmology, Lund University, Lund, Sweden

\* karl-johan.hellgren@regionvarmland.se

## Abstract

### Background

A better characterization of diabetic retinopathy (DR) may be helpful to monitor early disease, predict progression of DR, and to evaluate new treatment strategies. Visual function has been suggested to complement the assessment of microvascular lesions in DR but needs to be evaluated in longitudinal studies.

### Objectives

This prospective longitudinal cohort study investigated whether early visual field deterioration in diabetes is associated with change in DR, and whether known risk factors as diabetes duration and glycated A1c (HbA1c) affect the visual field.

### Methods

People living with diabetes, 18 to 75 years of age, were consecutively recruited from the local DR screening program. Individuals with eye diseases other than DR that could affect the visual field, and those who had received previous local eye treatment for DR, could not be included. Participants who had completed a five-year follow-up were re-examined after nine and ten years from baseline. The most important outcome was deterioration in series of visual fields as determined by an experimental model tailored for people living with diabetes. Stages of DR were evaluated according to the Early Treatment Diabetic Retinopathy Study (ETDRS) scale, and glycemic control by measurement of HbA1c.

### Results

Fifty-six participants (median age 69 years at the last visit, 35 males) completed 608 out of 616 scheduled visits during ten years of follow-up. Progression and regression of DR occurred most often between no (ETDRS level 10) and minimal (ETDRS level 20) DR. The number of deteriorated test points increased annually by 11% (95% CI: 6.9–15.3) and were not associated with change in DR but with higher levels of HbA1c.

**Data availability statement:** The data cannot be shared publicly because it contains extensive amount of sensitive personal data on a relatively small number of individuals from a specified and local area which also can be used to identify people, e.g. age, sex, diabetes type, diabetes duration, repeated biometric measurements and updates on diseases and medications. According to the ethical approvals from 2004, 2006 and 2015, data access is currently limited to participating researchers. Researchers may request deidentified data from the data owner, County Council of Värmland (Region Värmland) https://regionvarmland.se, datauttagforskning@ regionvarmland.se or contact the corresponding author for guidance. Access may require an approved application from the Swedish Ethical Review Authority (https://etikprovningsmyn-digheten.se).

**Funding:** The funders had no role in study design, data collection and analysis, decision to publish, or preparation of the manu-script. The funders were: The Department of Ophthalmology, Karlstad Hospital and Centre for Clinical Research, Region Värmland County, Sweden. ALF-funding Region Örebro County, Sweden. The Swedish Eye Foundation, Ögonfonden.

**Competing interests:** I have read the journal's policy and the authors of this manuscript have the following competing interests: Hellgren KJ has no competing interests. Bengtsson B receives royalties from Carl Zeiss Meditec and has been a consultant of Carl Zeiss Meditec. This does not alter our adherence to PLOS ONE policies on sharing data and materials.

## Conclusions

Early deterioration of visual function occurred independently of DR and was associated with worse glycemic control, suggesting that the metabolic disturbances due to diabetes induced a primary deterioration of sensitivity in the visual field.

## Introduction

Worldwide 537 million people were estimated to have diabetes in 2021, corresponding to a prevalence rate of 10.5% [1]. In Sweden the prevalence rate is estimated to reach above 10% in 2050 [2]. Diabetic retinopathy (DR) is a common retinal microvascular complication to diabetes with a global prevalence of 22% [3]. In Sweden DR is present in approximately 1/3 of people with diabetes [4,5]. Controlling blood glucose and blood pressure can postpone DR and local eye treatment with intravitreal injections, laser and surgery can reduce blindness when DR has become sight-threatening. Knowledge about how diabetes affects the retina at an early stage could potentially be useful to better monitor DR and develop new treatment strategies aiming to prevent the later sight-threatening stages [6,7].

Retinal neural function, in addition to microvascular lesions, has been proposed to be useful in a revised multidimensional diabetic retinal disease severity scale [8]. Retinal neural function is typically quantified by visual function tests, e.g., contrast sensitivity, color vision, and visual fields, which have indicated retinal function to be affected at early stages of DR [9]. Visual field tests have the advantage of quantifying visual function at several retinal locations. The most widely used method to assess and monitor visual fields is standard automated perimetry (SAP).

Cross-sectional studies have reported SAP to be affected by DR, and more so with more severe DR [10–12]. Though, these studies did not find any difference between eyes with no and mild DR, but such differences have been reported in a few other studies [13–15]. Some studies also considered whether risk factors for DR, as diabetes duration and glycemic control, were associated with SAP, but results were inconclusive [10,13,16,17]. However, longitudinal studies are compulsory to investigate whether subtle changes, deterioration or improvement, in visual function correlate or precede microvascular damage. Such subtle changes are difficult to detect by only arbitrary examine graphic test results as, e.g., grey scales of raw threshold values.

We have previously published the first part of our longitudinal study, which aimed at investigate whether subtle change in the visual field could be discerned at different sever-ity levels of DR, by using a new experimental model flagging significantly deteriorated or improved test points, further described in the method section [18]. After the five-years' follow-up, subtle deterioration had occurred according to the model for detection of significant change in eyes with stable no DR or mild DR. Change in DR occurred only in very few eyes, therefore the correlation between visual field deterioration and pro-gression/regression of DR could not be investigated. Since this previous report we have fortified the significance limits for detection of subtle visual field change in eyes with no or mild DR.

The primary aim of the current report was to investigate whether early visual field change in diabetes is associated with change in DR. The secondary aim was to investigate whether known risk factors for progression of DR, such as diabetes duration and HbA1c, also affect early change in function. To that end, participants from the previous five-year follow-up study were invited to two additional follow-up visits at nine and ten years from baseline.

## Methods

### Study setting

The study adhered to the declaration of Helsinki and was approved by the Swedish Ethical Review Authority´s regional board in Lund and Uppsala. All participants gave both oral and written informed consent. The study was conducted at the Department of Ophthalmology, Karlstad Hospital, Sweden between 2006 – 2019.

### Population

The participants were consecutively recruited from the local screening program for DR at the Department of Ophthalmology, Karlstad Hospital, Sweden, between September 10, 2006, and May 7, 2009. Eligibility criteria were people living with diabetes aged 18 to 75 years, who attended the screening program for DR in the county of Värmland, for more details see Hellgren et al. 2013 [19]. The current report applies to participants who completed five years of follow-up and continued in the extension of prospective data collection with nine- and ten-year follow-up between October 2015 and December 2019.

### Study design

A prospective longitudinal cohort study. The participants were initially scheduled for study visits every sixth months for the first three years and annually for another two years until five years from baseline, and then in the present continuation at nine and ten years from baseline. All visits assessed visual function by SAP and visual acuity, DR, and measurements of HbA1c level and blood pressure. Participants diagnosed with diabetes at age ≤ 30 years were defined as having early-onset diabetes.

### Measures

Diabetic retinopathy was graded according to the ETDRS severity scale at each follow-up visit [20]. A single grader (KJH) masked for participant identity performed all gradings. Seven field 35° stereographic digital fundus photographs were assessed by a Topcon Retinal Camera 50 DX (Topcon Corp., Tokyo, Japan).

Blood samples for measurement of HbA1c were analyzed using a TOSOH G8 Analyzer (Medinor, Tokyo, Japan), normal range 27 to 42 mmol/mol, < 50 years of age, and 31 to 46 mmol/mol for ≥ 50 years of age. Blood pressure was measured with the participant in a sitting position and calculated as the mean of two measurements.

Best corrected visual acuity was measured at 4 meters distance with an Early Treatment Diabetic Retinopathy Study (ETDRS) chart.

Ocular and medical history was updated at each visit. If the updated history or examination indicated a visual field change by causes other than diabetes, the participant was examined by an ophthalmologist, including a slit lamp examination and ophthalmoscopy.

Visual fields were tested using the SITA Standard 24-2 test program implemented in the Humphrey Field Analyzer (HFA) 750 (Carl Zeiss Meditec Inc, Dublin, CA, USA).

We have previously designed an experimental model for detection of visual field change, deterioration, and improvement, similar to the model commonly used for detection of visual field deterioration caused by glaucoma. Our model was based on a database of test-retest variability of age corrected light sensitivity threshold values in an independent sample of 50 people living with diabetes and a wide range of DR as graded by the ETDRS severity scale from Level 10 to Level 75 [12]. With knowledge of such random test-retest variability, the model identifies change, deterioration, or improvement, by comparing each follow-up visual

field test result with that obtained at baseline and flag test points which changed significantly at the p < 0.05 level. The baseline visual field was represented by an average of the first two tests obtained during the study period. Thus, in the present study change in visual fields was obtained from nine visits, at 1, 1 ½, 2, 2 ½, 3, 4, 5, 9 and 10 years from baseline. We expected our model to be more sensitive for detection of change than the conventional "single field analysis" implemented in the HFA comparing test results to age-corrected normal threshold data. The early version of this model was used to identify deteriorated and improved test points and was described and evaluated in previous reports [18,19]. The significance limits for change used in the present report were fortified to ameliorate detection of early change, by including additional 30 people living with diabetes with no or mild DR to the test-retest data-base. The additional data also enabled to account for global defect depth in addition to local defect depth and a more precise consideration of test point location [21].

In our previous studies defining the limits for significant change and in the present longitudinal study, eyes with previous local eye treatment for DR, either with photocoagulation or intravitreal injections, were excluded, and so were individuals with any other disease known to possibly affect the visual field, e.g., stroke, dementia, optic neuropathy, glaucoma and retinal vein occlusion. Reliability of visual fields was assessed by monitoring of participant gaze stability as observed by the perimetrist and by automated measurement of false positive responses required to ≤ 15% [22].

Outcomes for visual field change were number of test point locations showing significantly deteriorated and improved differential light sensitivity according to our fortified model for change. Outcomes for single visual fields were the age-adjusted perimetric summary index Mean Deviation (MD), an MD value below normal limit (p < 0.05), and the number of significantly depressed test points at the p < 0.05 level, all presented in the Statpac "Single Field Analysis." A DR change required a change of at least two steps on the ETDRS severity scale or if retinopathy level changed from level 10, no retinopathy, to level 20, microaneurysms only, or vice versa.

## Statistics

One randomly selected eye per participant was analyzed. Distributions of continuous data were skewed and subsequently presented as median and interquartile range (IQR). Change in participant characteristics and in conventional perimetric indices between baseline and the last visit were analyzed by Wilcoxon signed rank test for continuous and count data and McNemar's test for binary data. Statistical analysis was performed using IBM SPSS Statistics for Windows, Version 28.0 (IBM Corp, Armonk, NY). All included participants performed a reliable visual field at each follow-up visit.

A repeated measure analysis, general estimating equations (GEE), was applied to utilize all longitudinal data. To analyze perimetric outcomes change over time a linear model was applied for continuous data, i.e., change of MD values during follow-up, negative binomial models were applied for count data, i.e., the number of significantly depressed test points in the "single field analysis" and number of deteriorated/improved test points using the experimental model for change. The metric of these binomial models is the incidence rate ratio (IRR), which expresses the change in counts of depressed and deteriorated/improved test points per unit of the independent variable, e.g., per year of follow-up. To analyze whether deteriorated test points were associated with DR change, we applied a logistic model, controlling for the effect of time. Factors potentially associated with number of deteriorated test points were those considered as established risk factors for progression of DR, i.e., diabetes duration, HbA1c, and blood pressure [23]. Association between potential factors and deteriorated test points was analyzed by binomial models controlling for the effect of time. Thus, the

model included all values of the dependent variable, deteriorated test points, and all values of the potential predictors from each follow-up visit, e.g., HbA1c values at 1, 1 ½ , 2, 2 ½ , 3, 4, 5, 9 and 10 years. A multivariable model included the potential factors mentioned above.

## Results

The recruitment for the nine- and ten-year continuation included 56 of 63 participants who had completed five years of follow-up. Of those seven who did not participate, three declined further participation, two were severely ill and could not participate, one had died, and one had received photocoagulation in the study eye, From the baseline visit, during the entire follow-up, until the ten-year visit, the 56 participants had each eleven scheduled visits. In total the participants completed 608/616 visits.

Thirty-five of 56 (63%) participants were men, median age at the last visit was 69 (IQR 61–75) years. Nine had early-onset diabetes. At the last visit compared to baseline, diastolic blood pressure was lower, 75 vs 80 mmHg, p = 0.009, and a larger proportion had antihypertensive treatment: 39/56 vs 28/56, p < 0.001, Table 1.

Most eyes had no DR, ETDRS level 10, or microaneurysms only, ETDRS level 20, both at baseline and at the last visit, 44/56 (79%) and 38/56 (68%) respectively, while higher levels of retinopathy were less common, (Fig 1). Eleven eyes had DR change at the last visit, of which there was progression in eight eyes and regression in three eyes. Out of all follow-up visits during the ten-year period progression of DR occurred in total at 24 visits in 12 eyes. The progressions occurred most often from ETDRS level 10, no DR, to level 20, microaneurysms only, 15/24 (63%). Out of all follow-up visits during the ten-year period regression of DR occurred in total at 29 visits in nine eyes. The regressions occurred most often from ETDRS level 20 to level 10, 27/29 (93%). Three study eyes, which had the worse levels of DR at the last visit, ETDRS level ≥ 43, showed progression from baseline, from level 20 to level 43, from level 35 to 53, and from level 35 to 61, respectively.

## Visual fields

The visual field change analysis identified an increasing number of deteriorated test points over time. Deteriorated test points increased in average by IRR 1.110 (95% CI: 1.069–1.153), thus an annual increase of 11.0%. The individual highest number of deteriorated test points

**Table 1. Participant characteristics at baseline and at last visit (all participants who fulfilled the nine/ten-year follow-up).**

|  | Baseline (n = 56) | Last visit (n = 56) |
|---|---|---|
|  | median (IQR) | median (IQR) |
| Age, years | 59(52–65) | 69(61–75) |
| Diabetes duration, years | 10(4–22) | 20(14–31) |
| HbA1c, mmol/mol | 58(52–67) | 61(54–69) |
| Systolic BP, mm Hg | 130(125–145) | 140(125–145) |
| Diastolic BP, mm Hg | 80(70–85) | 75(66–80) |
| Hypertensive treatment n (%) | 29(52) | 39(70) |
| Diabetes treatment n (%) |  |  |
| Diet only | 7(12) | 2(4) |
| Non-insulin only hypoglycemic agents only | 14(25) | 13(23) |
| Non-insulin hypoglycemic agents and insulin | 12(21) | 19(34) |
| Insulin only | 23(41) | 22(39) |

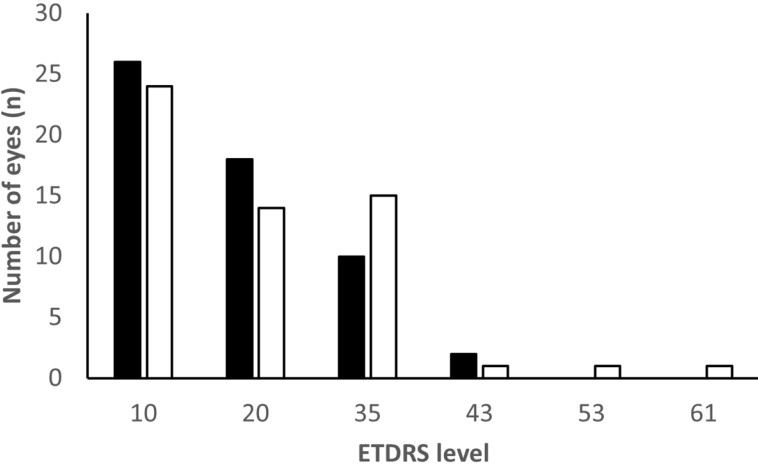

**Fig 1. Early Treatment Diabetic Retinopathy Study (ETDRS) levels at the first and the final visit.** Most eyes had no vascular diabetic retinopathy, ETDRS level 10, or mild diabetic retinopathy, ETDRS levels 20–35, at baseline (black bars) and at the last visit (white bars). One eye had proliferative diabetic retinopathy, ETDRS level 61, at the last visit.

occurred more often at longer follow-up, in 6%, 16%, and 34% of eyes at one, five, and ten years of follow-up respectively. The significant increase of deteriorated test points was sustained after excluding the three eyes which had progressed to moderate or worse DR, ETDRS level ≥ 43, at the last visit. The number of improved test points remained at a stable low level, 1.013 (0.974–1.055).

In most participants the conventional perimetric indices at baseline, as well as at the last visit, were within normal limits, but the median MD value at baseline decreased from -0.38 dB (IQR -1.20–0.34 dB) to -0.90 dB (IQR -1.79–0.36), p = 0.03. In the repeated measure analysis, MD value deteriorated by 0.08 dB/year (95% CI: 0.03–0.12). The number of significantly depressed test points increased by an IRR of 1.063 (95% CI: 1.033–1.094), thus an annual increase of 6.3% depressed test points per year. After excluding the three eyes which at the last visit had developed moderate or worse DR, ETDRS level ≥ 43, the differences in MD were still significant and number of depressed test points had increased significantly, p < 0.001.

Number of deteriorated test points was not significantly associated with a change in DR, odds ratio (OR) 1.015 (95% CI: 0.957–1.077), nor with progression of DR or regression of DR, OR 1.013 (0.939–1.093) and 1.015 (0.938–1.098) respectively. Of the three eyes with the highest level of retinopathy at the last visit, the two eyes which progressed to levels 43 and 53 had only 3 and 0 deteriorated test points respectively. The eye which progressed to proliferative DR, level 61, had 14 deteriorated test points at the last visit.

Our analysis revealed that higher HbA1c and lower diastolic blood pressure were significantly associated with more deteriorated test points; IRR were 1.024 (95% CI: 1.007–1.042) and 0.979 (0.960–0.999), respectively, Table 2. Thus, an average increase of 2.4% (0.7 – 4.2) deteriorated test points for each mmol/mol higher HbA1c and an average decrease of 2.1% (0.1 – 4.0) deteriorated test points for each mmHg higher blood pressure. A significant interaction between diabetes duration and HbA1c effect was identified. The multivariable model which aimed at identifying independent risk factors for deterioration included diabetes duration, HbA1c, the pre-identified significant interaction effect between diabetes duration and HbA1c, diastolic blood pressure, and adjustment for follow-up, Table 3. In this multivariable model the effect of HbA1c on the number of deteriorated test points was moderated by diabetes duration (Fig 2A–D). A higher number of deteriorated test points was associated with

Table 2. Factors associated with number of deteriorated test points, adjusted for time (years in study).

|  | IRR | 95% CI | p |
|---|---|---|---|
| Diabetes duration, per year | 1.017 | 0.998–1.037 | 0.081 |
| HbA1c, per mmol/mol | 1.024 | 1.007–1.042 | 0.006 |
| Systolic blood pressure, per mmHg | 0.996 | 0.982–1.009 | 0.538 |
| Diastolic blood pressure, per mmHg | 0.979 | 0.960–0.999 | 0.036 |

Table 3. Factors associated with number of deteriorated test points, multivariable model.

|  | IRR | 95% CI | p |
|---|---|---|---|
| Diabetes duration, per year | 1.106 | 1.012–1.208 | 0.027 |
| HbA1c, per mmol/mol | 1.043 | 1.005–1.081 | 0.025 |
| Diabetes duration x HbA1c | 0.998 | 0.997–1.000 | 0.048 |
| Diastolic BP, per mmHg | 0.985 | 0.966–1.004 | 0.116 |
| Time, per year in study | 1.097 | 1.061–1.134 | < 0.001 |

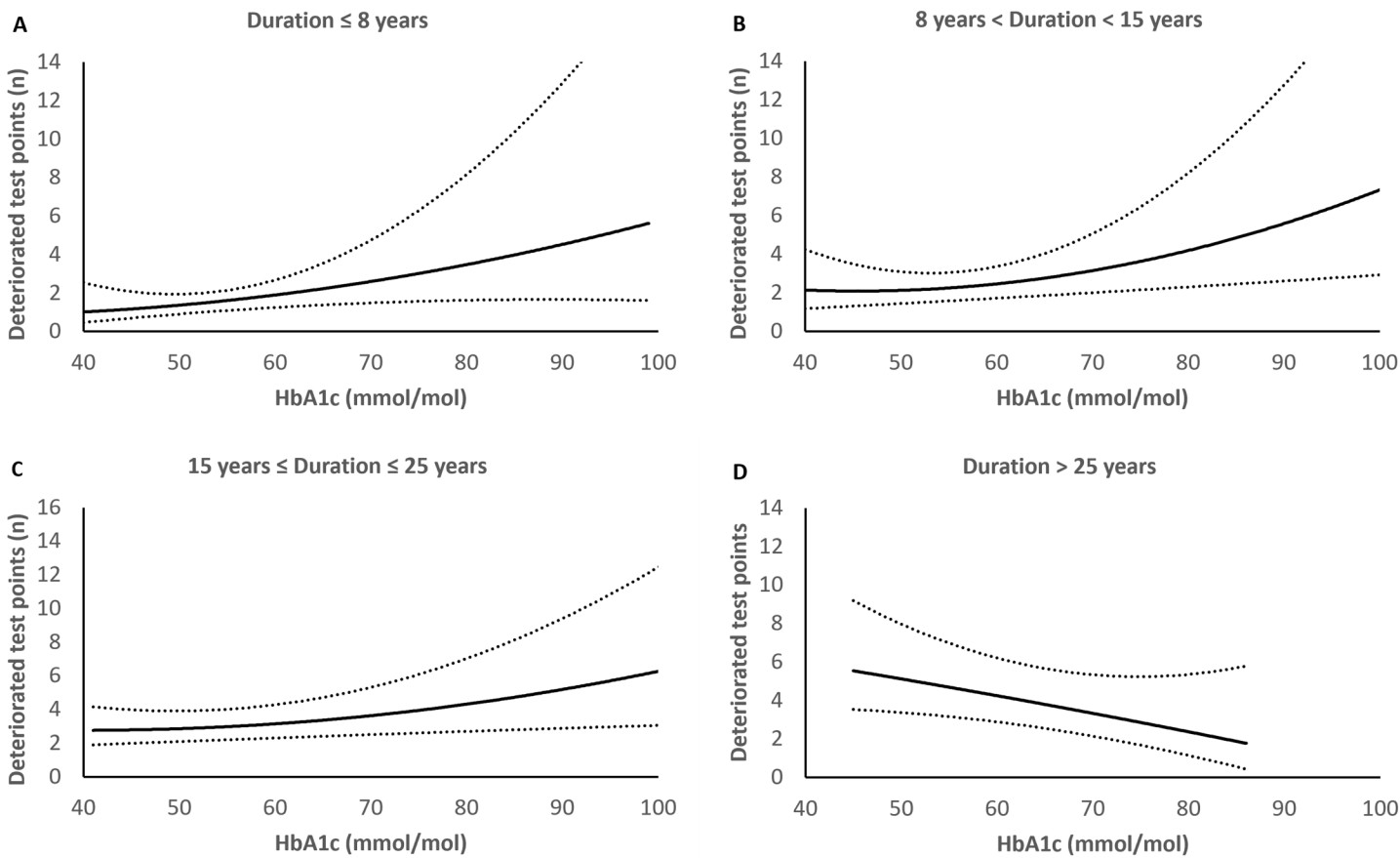

**Fig 2. A–D. Number of deteriorated test points with 95% CI (dotted lines) by glycated hemoglobin A1c (HbA1c), from the first (A) to fourth (D) quartile of diabetes duration.** The effect of HbA1c on deteriorated test points varied with diabetes duration. Higher HbA1c was associated with a higher number of deteriorated test points except for those with the longest duration, where higher HbA1c was associated with a lower number of deteriorated test points.

higher HbA1c except at long duration beyond 25 years, where a higher number of deteriorated test points was associated with lower HbA1c.

Visual acuity decreased by 0.336 (0.171–0.502) letters/year. Visual acuity change was not associated with number of perimetrically deteriorated test points; the IRR was 1.019 (0.963–1.079) for each letter of visual acuity loss. Nine eyes had undergone cataract surgery during the ten-year follow-up. The results were similar when these cases were excluded.

## Discussion

With 10 years of follow-up this study explored visual field change in people living with diabetes. An exceptional long follow-up in clinical research but a relevant time frame in the perspective of people with a chronic disease that may cause late debilitating complications such as blindness caused by DR. The novelty of the study, in addition to the long follow-up, is that no previous study has been designed to investigate early change in visual function in diabetes which was assessed by a model tailored for diabetes to detect deteriorations and improvements of the visual field. The aim was to explore an association between change in visual function and DR, and to investigate possible risk factors of importance for such early change in visual function. We hypothesized that visual field deterioration could predict progression of DR and that factors important for microvascular diabetes complications, such as diabetes duration and glycemic control, would also be important for early visual field change. Our results revealed subtle but significant deterioration of the visual field over time, but no direct nor delayed association between the subtle visual field changes and DR change. Hemoglobin A1c levels were associated with visual field deterioration, but the effect was moderated by duration of diabetes. A higher HbA1c was associated with a higher number of deteriorated test points but at long diabetes duration, > 25 years, a higher HbA1c was associated with a lower number of deteriorated test points.

Previous studies have indicated that visual function may predict incident DR but did not analyze change in visual function [24,25]. However, a subgroup analysis of the Eurocondor study revealed deteriorated function in early diabetes using multifocal electroretinography [26]. Cross-sectional studies have reported visual field depression in eyes with no and mild DR but with inconclusive results [10,11,13–15]. Worse glycemic control has been suggested to be associated with visual field defects in a large sample (n = 530) without DR using supra threshold perimetry [27]. In line with these results, we observed that HbA1c affected the number of deteriorated test points.

The present study does not support the idea of a correlation between visual function and early stages of DR but revealed glycemic control to be associated with deteriorations in the visual field. The effect of glycemic control varied with the length of diabetes duration. At short and medium–long diabetes duration, up to 25 years, a higher HbA1c level was associated with a higher number of deteriorated test points. The result suggests that the neural retina is affected by metabolic disturbances due to diabetes. However, at longer duration the correlation shifted, in that a higher HbA1c level was associated with a lower number of deteriorated test points. Reasons for this negative correlation could only be speculated upon. Possibly the neuronal retina is less responsive to worse metabolic control when duration of diabetes is long, which may truncate the detection of deteriorations in eyes with long duration and high HbA1c. People living with diabetes with long diabetes duration may also differ due to survival bias, having a subgroup of diabetes with higher resilience to complications [28]. It may also be possible that tight glycemic control, especially with long duration, would reflect a higher risk of hypoglycemia, which has been associated with visual dysfunction [29,30].

Our experimental model for change almost doubled the rate of deteriorated test points compared to the conventional "single field analysis" rate of significantly depressed test points defined by comparison to age-corrected normal threshold values: 11% vs 6% respectively.

Thus, in analogy to the progression analysis used for monitoring glaucoma, our model for change seemed to be an effective strategy for monitoring of early retinal neuronal dysfunction in diabetes [31]. The yearly increase in number of deteriorated test points may seem high, but it is important to consider that the model is very sensitive in detecting subtle change, especially when being close to normal, as was the case for the participants in the present study. Thus, the detected deterioration of function is unlikely to be symptomatic and does not affect activities in daily life. However, the results have implications for future research. A model such as ours, detecting deterioration of sensitivity in the visual field, provides opportunities to better characterize diabetic eye disease in advance of visible DR. Retinal neurodegeneration has been described to occur early in diabetes with change in thickness of retinal layers [32,33]. It would be possible to investigate whether early degeneration is predicted by or associated with loss of function in the visual field.

The strengths of the study are the long follow-up; highly adherent participants completing 99% (606/618) of visits; consistent assessment of measures of visual function, guaranteed by one examiner and assessment of seven field photographs by trained photographers; and masking study identity and visual field results at grading of retinopathy according to the gold standard for assessing DR change. We do acknowledge that the study has limitations. The participants were relatively few as were follow-up visits at the end of the ten-year period. Photographs during the first five years were graded on color slide film while the nine- and ten-year visits relied on digital color photographs. However, color slide film and digital color photographs have been reported to be comparable using the seven ETDRS standard fields, as was the case for the present study [34]. We do not believe our results were driven by the few eyes with progression to moderate or worse DR because the deterioration of sensitivity in visual fields remained after excluding these participants from the analysis. Furthermore, because the detected DR change occurred most often between the levels of no and mild DR, we could not analyze a possible later association between the early subtle visual field deterioration and later progression to severe non-proliferative and proliferative DR.

## Conclusion

A subtle deterioration in visual function was seen over time in eyes with no and mild DR. This deterioration did not correlate to a microvascular change in DR levels, but the deterioration was more pronounced with worse glycemic control except at long duration. The result suggests a primary neuronal dysfunction due to diabetes. A sensitive model like ours to detect change over time in visual function may be useful to monitor the first retinal response in diabetes. However, the consequences for development of sight-threatening DR and debilitating visual impairment is unclear.

AcknowledgmentWe thank Elisabet Agardh, MD, PhD, for valuable discussions regarding diabetic retinopathy and early retinal neurodegeneration in diabetes.

## Author contributions

**Conceptualization:** Karl-Johan Hellgren, Boel Bengtsson.

**Data curation:** Karl-Johan Hellgren.

**Formal analysis:** Karl-Johan Hellgren.

**Funding acquisition:** Karl-Johan Hellgren.

**Investigation:** Karl-Johan Hellgren.

**Methodology:** Karl-Johan Hellgren, Boel Bengtsson.

**Project administration:** Karl-Johan Hellgren.

Resources: Karl-Johan Hellgren.

Supervision: Boel Bengtsson.

Validation: Karl-Johan Hellgren.

Visualization: Karl-Johan Hellgren.

Writing – original draft: Karl-Johan Hellgren.

Writing – review & editing: Karl-Johan Hellgren, Boel Bengtsson.

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
