## [Decision Letter · Decision Letter 0]

27 Aug 2024

PONE-D-24-24590Ten Years of visual Field Change in Patients with Diabetes: A prospective longitudinal StudyPLOS ONE

Dear Dr. Hellgren,

Thank you for submitting your manuscript to PLOS ONE. After careful consideration, we feel that it has merit but does not fully meet PLOS ONE’s publication criteria as it currently stands. Therefore, we invite you to submit a revised version of the manuscript that addresses the points raised during the review process.

We look forward to receiving your revised manuscript.

Kind regards,

Daisuke Nagasato

Academic Editor

PLOS ONE

Journal Requirements:

When submitting your revision, we need you to address these additional requirements. 1. Please ensure that your manuscript meets PLOS ONE's style requirements, including those for file naming. The PLOS ONE style templates can be found at https://journals.plos.org/plosone/s/file?id=wjVg/PLOSOne_formatting_sample_main_body.pdf and https://journals.plos.org/plosone/s/file?id=ba62/PLOSOne_formatting_sample_title_authors_affiliations.pdf 2. Thank you for stating the following in the Competing Interests section: "I have read the journal's policy and the authors of this manuscript have the following competing interests: Hellgren KJ has no competing interests. Bengtsson B receives royalties from Carl Zeiss Meditec and has been a consultant of Carl Zeiss Meditec" Please confirm that this does not alter your adherence to all PLOS ONE policies on sharing data and materials, by including the following statement: ""This does not alter our adherence to  PLOS ONE policies on sharing data and materials.” (as detailed online in our guide for authors http://journals.plos.org/plosone/s/competing-interests).  If there are restrictions on sharing of data and/or materials, please state these. Please note that we cannot proceed with consideration of your article until this information has been declared.  Please include your updated Competing Interests statement in your cover letter; we will change the online submission form on your behalf. 3. In the online submission form, you indicated that "The data, which contains sensitive personal data, is not publicly available due to legal and ethical restrictions. Deidentified participant data is available for researchers after a reasonable request to the authors and the data owner, County Council of Värmland. Requests may be sent to the corresponding author, Karl-Johan Hellgren." All PLOS journals now require all data underlying the findings described in their manuscript to be freely available to other researchers, either 1. In a public repository, 2. Within the manuscript itself, or 3. Uploaded as supplementary information.This policy applies to all data except where public deposition would breach compliance with the protocol approved by your research ethics board. If your data cannot be made publicly available for ethical or legal reasons (e.g., public availability would compromise patient privacy), please explain your reasons on resubmission and your exemption request will be escalated for approval.

Reviewers' comments:

Reviewer's Responses to Questions

**Comments to the Author**

1. Is the manuscript technically sound, and do the data support the conclusions?

Reviewer #1: Partly

Reviewer #2: Yes

2. Has the statistical analysis been performed appropriately and rigorously?

Reviewer #1: Yes

Reviewer #2: Yes

3. Have the authors made all data underlying the findings in their manuscript fully available?

Reviewer #1: Yes

Reviewer #2: No

4. Is the manuscript presented in an intelligible fashion and written in standard English?

Reviewer #1: Yes

Reviewer #2: No

5. Review Comments to the Author

Reviewer #1: This paper describes the details of visual field changes in diabetic patients in a long-term observation. While well-written in total, there remain some points that might need to be clarified:

1. Line 108-109: The authors state about the definition of type 1 diabetes. However, the diagnosis of type 1 diabetes should be based on internal medicine criteria. It is not uncommon for individuals under 30 to be diagnosed with type 2 diabetes and subsequently require insulin therapy. If it is unclear whether patients have type 1 or type 2 diabetes, and if this distinction is not a crucial factor in the context of your study, it is advisable to state this uncertainty explicitly.

2. The authors position the reliability of their analysis based on the finding that excluding the three eyes with the poorest final visual acuity and moderate or worse DR did not alter the results. However, is there a rationale for excluding these three eyes? Additionally, why were only cases with ETDRS levels of 43 or higher excluded? If there is previous literature or a basis for this decision, it should be stated.

3. In Figure 2, the authors describe the correlation between HbA1c levels and the number of visual field deterioration points across different durations of diabetes. At what point in time were these HbA1c values measured? Did you consider the fluctuations in HbA1c levels? If so, please discuss this in the manuscript.

4. Line 259: It is stated that cataracts or cataract surgery affected the visual field test results in two cases. Were these two cases excluded from the analysis? If they were not, what is the reason?

5. The finding that there is a negative correlation between HbA1c levels and the number of visual field deterioration points in the group with a longer duration of diabetes is indeed very interesting. However, could it be that cases with a longer duration of diabetes and poor HbA1c control inherently have worse visual field impairment compared to other cases? In other words, is it possible that the visual field impairment due to diabetes has already reached its peak in these cases? If you wish to assert, as mentioned on line 270, that 'a higher HbA1c was associated with a lower number of deteriorated test points,' it would first be necessary to examine whether there is a significant difference in baseline visual field impairment between the groups stratified by duration of diabetes.

Reviewer #2: Thank you very much for invitation, It is an important area of diabetic care, but the duration of the data collection was so long which was 2019, the lowers the quality of the study

Reviewer #1 comments

Title: Ten Years of visual Field Change in Patients with Diabetes: A prospective longitudinal Study-better edited as: Ten years of visual field change in patients with diabetes: A prospective longitudinal study

Abstract

The abstract must fit with Plos one guideline

Introduction

Included the magnitude of diabetes, and DR globally and in your country as well

The relationship between Visual field function and DR change must be clearly stated.

Method

What was your eligibility criteria?

The selection criteria is not clear? How do you reassure that the participants do not have a visual change at baseline? When does the baseline assessment done?

DR is the most common microvascular complication in T1DM, but less likely in T2DM? but most of your participants were T2DM, why?

Line 123: similar to the that….editing

What was the sample size including sample size calculation

What was the sampling procedure?

There must be a clear operational definition of Visual field change and DR

A multivariable model included all potential factors looks incomplete please rewrite it.

Better rewrite the method parts section by section. As study setting, design, population (including and excluding), sample size and sampling procedure, data collection tools and procedures, and method of analysis.

What was the data collection tools and procedures, State clearly in the method section

Who collect the data?

How do you control a visual field change by other causes?

Why you prefer GEE other than LMM?

Result

Of those eight who did not participate this is not correctly calculated

Line 180….From line….from

Table 1: HbA1c, why do not use %?

Diabetes treatment (n) four of them under this category lacks clarity. So try to rewrite it clearly.

Line 190-there is controversies 79% Vs 68%,

Line 197, three study eyes not clear

Line 205: needs subtitle as Visual field ……..

207 IRR….describe it as long form with short form then proceed with short form

Line 223-225, Is it COR or AOR? Is it recommended in this study?

Make a consistent description……in DR

Line 230-232; The number 2.4 % and 2.1% are not correct based on the CIs stated.

Discussion

Lacks novelty need to be rewrite it

Start with the aim of the study, then proceed to your hypothesis

Use consistent referencing line 281 and 288

Please include the study implications and recommendations,

Conclusion better stated separately with the discussion and stick to your findings

The entire manuscript need to be edited by professional language expert

6. PLOS authors have the option to publish the peer review history of their article (what does this mean? ). If published, this will include your full peer review and any attached files.

**Do you want your identity to be public for this peer review?** For information about this choice, including consent withdrawal, please see our Privacy Policy .

Reviewer #1: No

Reviewer #2: **Yes: ** Abere Woretaw Azagew

---

## [Author Response · Author response to Decision Letter 1]

7 Nov 2024

We thank the editor and the reviewers for the comments. The thorough critical reviews are highly appreciated. Please, se our replies below.

Yours sincerely,

Karl-Johan Hellgren, MD, PhD

Comments and replies to Academic Editor Daisuke Nagasato

Reply: We have ensured that the manuscript meets the requirements.

-------------------

"I have read the journal's policy and the authors of this manuscript have the following competing interests: Hellgren KJ has no competing interests. Bengtsson B receives royalties from Carl Zeiss Meditec and has been a consultant of Carl Zeiss Meditec"

Reply: The Competing Interest statement does not alter our adherence PLOS ONE policies and the statement have changed as proposed and included in the cover letter:

“I have read the journal's policy and the authors of this manuscript have the following competing interests: Hellgren KJ has no competing interests. Bengtsson B receives royalties from Carl Zeiss Meditec and has been a consultant of Carl Zeiss Meditec. This does not alter our adherence to PLOS ONE policies on sharing data and materials.”

-------------------

3. In the online submission form, you indicated that "The data, which contains sensitive personal data, is not publicly available due to legal and ethical restrictions. Deidentified participant data is available for researchers after a reasonable request to the authors and the data owner, County Council of Värmland. Requests may be sent to the corresponding author, Karl-Johan Hellgren."

Reply: The data cannot be shared publicly because it contains extensive amount of sensitive personal data on a relatively small number of individuals from a specified and local area which also can be used to identify people, e.g. age, sex, diabetes type, diabetes duration, repeated biometric measurements and updates on diseases and medications. According to the ethical approvals from 2004, 2006 and 2015, data access is currently limited to participating researchers. Researchers may request deidentified data from the data owner, County Council of Värmland (Region Värmland) https://regionvarmland.se, datauttagforskning@regionvarmland.se or contact the corresponding author for guidance. Access may require an approved application from the Swedish Ethical Review Authority (https://etikprovningsmyndigheten.se).

-------------------

Comments and replies to Reviewer #1

1. Line 108-109: The authors state about the definition of type 1 diabetes. However, the diagnosis of type 1 diabetes should be based on internal medicine criteria. It is not uncommon for individuals under 30 to be diagnosed with type 2 diabetes and subsequently require insulin therapy. If it is unclear whether patients have type 1 or type 2 diabetes, and if this distinction is not a crucial factor in the context of your study, it is advisable to state this uncertainty explicitly.

Reply: We do agree that our definition of diabetes type may imply some uncertainty in proportion of diabetes type in patient characteristics, but this uncertainty did not affect any other results since diabetes type was not included in any of the statistical models and thus not an outcome variable. A review of medical records to possibly confirm internal medicine criteria at diagnosis as type 1 or type 2 was not performed in this prospective study, since most patients were diagnosed with diabetes years before the start of the current study.

-------------------

2. The authors position the reliability of their analysis based on the finding that excluding the three eyes with the poorest final visual acuity and moderate or worse DR did not alter the results. However, is there a rationale for excluding these three eyes? Additionally, why were only cases with ETDRS levels of 43 or higher excluded? If there is previous literature or a basis for this decision, it should be stated.

Reply: No association was seen between visual field deterioration and DR change. Most DR change occurred between no to mild DR and vice versa, thus the result indicates a visual dysfunction to occur early in a microvascular perspective. Three eyes progressed to level 43, moderate DR, or worse. An additional analysis omitting these eyes were conducted to ensure that the results were not driven by these eyes. A clarification has been added to the Result section, subheading Visual fields, Revised Manuscript with Track Changes, line 254 – 255.

-------------------

3. In Figure 2, the authors describe the correlation between HbA1c levels and the number of visual field deterioration points across different durations of diabetes. At what point in time were these HbA1c values measured? Did you consider the fluctuations in HbA1c levels? If so, please discuss this in the manuscript.

Reply: The model included all HbA1c from each follow-up visit, i.e. 1, 1 ½, 2, 2 ½, 3, 4, 5, 9 and 10 years from baseline. All patient series of HbA1c values was included in the analysis. A clarification has been inserted in the Method section, subheading Statistics, Revised Manuscript with Track Changes, Line 212 - 214.

“Thus, the model included all values of the outcome variable, i.e. number of deteriorated test points, and all measures of the potential predictors from each follow-up visit, e.g. HbA1c values at 1, 1 ½, 2, 2 ½, 3, 4, 5, 9 and 10 years. “

-------------------

4. Line 259: It is stated that cataracts or cataract surgery affected the visual field test results in two cases. Were these two cases excluded from the analysis? If they were not, what is the reason?

Reply: No, the two cases were not excluded, but a rerun of the analysis excluding these cases revealed a similar result (deteriorated test points increased by IRR 1.118 (1.077 – 1.161) compared to 1.110 (1.069 – 1.153)). Neither did the result change in any meaningful way if all nine subjects which had cataract surgery during follow-up were excluded from the analysis (deteriorated test points increased by IRR 1.114 (1.068 – 1.162).

A clarification has been made and by rephrasing the sentence, Result section, Manuscript with Track Changes, line 303 – 304.

From “There was no effect of cataract or cataract surgery on the visual field test results in seven of the nine eyes.”

To “The results were similar when these cases were excluded.”

-------------------

5. The finding that there is a negative correlation between HbA1c levels and the number of visual field deterioration points in the group with a longer duration of diabetes is indeed very interesting. However, could it be that cases with a longer duration of diabetes and poor HbA1c control inherently have worse visual field impairment compared to other cases? In other words, is it possible that the visual field impairment due to diabetes has already reached its peak in these cases? If you wish to assert, as mentioned on line 270, that 'a higher HbA1c was associated with a lower number of deteriorated test points,' it would first be necessary to examine whether there is a significant difference in baseline visual field impairment between the groups stratified by duration of diabetes.

Reply: The analysis of risk factors for the early deterioration found a slower deterioration rate with worse HbA1c in cases with long duration. Thus, the deterioration did not seize. We therefore do not think it is possible that the impairment has reached a peak. We have not stratified the baseline data because it renders too few cases for a statistical inference analysis, e.g. only 11/56 cases had diabetes duration > 25 years at baseline.

-------------------

Comments and replies to Reviewer #2

Title: Ten Years of visual Field Change in Patients with Diabetes: A prospective longitudinal Study-better edited as: Ten years of visual field change in patients with diabetes: A prospective longitudinal study

Reply: Capital letters have been changed to lowercase letters as suggested.

-------------------

Abstract

The abstract must fit with Plos one guideline

Reply: The abstract has been changed accordingly.

-------------------

Introduction

Included the magnitude of diabetes, and DR globally and in your country as well

Reply: A new paragraph to set the scene has been added to the Introduction, Manuscript with Track Changes, line 75 - 83:

“Worldwide 537 million people were estimated to have diabetes in 2021 corresponding to a prevalence rate of 10.5 % [1]. In Sweden the prevalence rate is estimated to reach above 10% in 2050 [2]. Diabetic retinopathy is a common retinal microvascular complication to diabetes with a global prevalence of 22 % [3]. In Sweden DR is present in approximately 1/3 of people with diabetes [4,5]. Controlling blood glucose and blood pressure can postpone DR and local eye treatment with intravitreal injections, laser and surgery can reduce blindness when DR has become sight-threatening. Knowledge how diabetes affects the retina at an early stage could potentially be useful to better monitor DR and develop new treatment strategies aiming to prevent the later sight-threatening stages [6,7].”

And the new references have been added to References:

1. Sun H, Saeedi P, Karuranga S, Pinkepank M, Ogurtsova K, Duncan BB, et al. IDF Diabetes Atlas: Global, regional and country-level diabetes prevalence estimates for 2021 and projections for 2045. Diabetes Res Clin Pract. 2022;183:109119. doi: 10.1016/j.diabres.2021.109119

2. Andersson T, Ahlbom A, Carlsson S. Diabetes Prevalence in Sweden at Present and Projections for Year 2050. PLoS One. 2015;10(11):e0143084. doi:10.1371/journal.pone.0143084

3. Teo ZL, Tham YC, Yu M, Chee ML, Rim TH, Cheung N, et al. Global Prevalence of Diabetic Retinopathy and Projection of Burden through 2045: Systematic Review and Meta-analysis. Ophthalmology. 2021;128(11):1580-1591. doi:10.1016/j.ophtha.2021.04.027

4. Olafsdottir E, Andersson DK, Dedorsson I, Stefánsson E. The prevalence of retinopathy in subjects with and without type 2 diabetes mellitus. Acta Ophthalmol. 2014;92(2):133-7. doi:10.1111/aos.12095

5. Sharif A, Smith DR, Hellgren KJ, Jendle J. Diabetic retinopathy among the elderly with type 2 diabetes: A Nationwide longitudinal registry study. Acta Ophthalmol. 2024;102(6):e883-e892. doi:10.1111/aos.16659

6. Antonetti DA, Klein R, Gardner TW. Diabetic retinopathy. N Engl J Med. 2012;366(13):1227-39. doi:10.1056/NEJMra1005073

7. Stitt AW, Curtis TM, Chen M, Medina RJ, McKay GJ, Jenkins A, et al. The progress in understanding and treatment of diabetic retinopathy. Prog Retin Eye Res. 2016;51:156-86. doi:10.1016/j.preteyeres.2015.08.001

-------------------

The relationship between Visual field function and DR change must be clearly stated.

Reply: A change (improvement or deterioration) in visual function and the development of DR can only be explored in longitudinal studies. The relationship has been further clarified in the introduction, Manuscript with Track Changes, line 91 – 101, especially at the end of the paragraph line 96 - 101.

From: “However, longitudinal studies are needed to investigate whether change in visual function correlates to change in DR, and what risk factors are important for such early change in function.”

To: “However, longitudinal studies are compulsory to investigate whether subtle changes deterioration or improvement in visual function, correlate or precede microvascular damage. Such subtle changes are difficult to detect by only arbitrary examine graphic test results e.g. grey scales of raw threshold values.”

-------------------

Method

What was your eligibility criteria?

Reply: A clarification has been made in the Methods section, Manuscript with Track Changes, line 127-129:

“Eligibility criteria were diabetic subjects aged 18 to 75 years, who attended the screening program for DR in the county of Värmland, for more details see Hellgren et al. 2013 [19].”

-------------------

The selection criteria is not clear?

Reply: The participants were consecutively recruited from the local screening program for DR with eligibility criteria above.

-------------------

How do you reassure that the participants do not have a visual change at baseline?

Reply: In this manuscript we consider visual field deterioration or improvement over time, i.e. change from the study start, which is not to the same as visual field change compared to a normal visual field. Visual field change due to diabetes may have occurred before inclusion in the study especially in participants with long diabetes duration.

A clarification has been inserted in the Introduction section, Manuscript with Track Changes, line 96 – 101:

From: “However, longitudinal studies are needed to investigate whether change in visual function correlates to change in DR, and what risk factors are important for such early change in function.”

To: “However, a longitudinal study design is compulsory to investigate whether subtle changes, deterioration or improvement, in visual function correlate or precede retinal microvascular damage. Such subtle changes are difficult to detect by only arbitrary examine graphic test results e.g. grey scales of raw threshold values.”

--------------

---

## [Decision Letter · Decision Letter 1]

26 Dec 2024

PONE-D-24-24590R1Ten years of visual field change in patients with diabetes: A prospective longitudinal studyPLOS ONE

Dear Dr. Hellgren,

Thank you for submitting your manuscript to PLOS ONE. After careful consideration, we feel that it has merit but does not fully meet PLOS ONE’s publication criteria as it currently stands. Therefore, we invite you to submit a revised version of the manuscript that addresses the points raised during the review process.

We look forward to receiving your revised manuscript.

Kind regards,

Daisuke Nagasato

Academic Editor

PLOS ONE

Journal Requirements:

Reviewers' comments:

Reviewer's Responses to Questions

**Comments to the Author**

1. If the authors have adequately addressed your comments raised in a previous round of review and you feel that this manuscript is now acceptable for publication, you may indicate that here to bypass the “Comments to the Author” section, enter your conflict of interest statement in the “Confidential to Editor” section, and submit your "Accept" recommendation.

Reviewer #1: (No Response)

Reviewer #2: All comments have been addressed

2. Is the manuscript technically sound, and do the data support the conclusions?

Reviewer #1: Yes

Reviewer #2: Yes

3. Has the statistical analysis been performed appropriately and rigorously?

Reviewer #1: Yes

Reviewer #2: Yes

4. Have the authors made all data underlying the findings in their manuscript fully available?

Reviewer #1: Yes

Reviewer #2: Yes

5. Is the manuscript presented in an intelligible fashion and written in standard English?

Reviewer #1: Yes

Reviewer #2: Yes

6. Review Comments to the Author

Reviewer #1: While the manuscript has shown improvement, I would like to provide some additional feedback for consideration.

1. If the distinction between type 1 and type 2 diabetes does not impact the results, it might not be necessary to separate them strictly. To avoid potential inaccuracies in the manuscript, defining individuals diagnosed with diabetes under the age of 30 as having 'early-onset diabetes' could be a clearer and more precise alternative.

2. Apologies for missing this in the initial review, but the title of Figure 1 appears to be just the legend, which seems very unusual. Perhaps the title could be something like 'ETDRS levels at the first visit and the final visit of the patients' to make it clearer.

Reviewer #2: Abstract

Purpose ….better change to Background /introduction based on the journal guideline. see the previous comments

Change the diabetes patients as “people living with diabetes”

The IQR should be stated in range form not the subtraction result, so edit the entire manuscript table

The study design must be expressed in the study design section

Include the study implication in the discussion section

Please use conclusion as separate a title after discussion.

7. PLOS authors have the option to publish the peer review history of their article (what does this mean? ). If published, this will include your full peer review and any attached files.

**Do you want your identity to be public for this peer review?** For information about this choice, including consent withdrawal, please see our Privacy Policy .

Reviewer #1: No

Reviewer #2: No

---

## [Author Response · Author response to Decision Letter 2]

21 Jan 2025

Thank you for the opportunity to improve the manuscript by additional changes. Please, se our replies below.

Yours sincerely,

Karl-Johan Hellgren, MD, PhD

Comments and replies

Journal Requirements:

Please review your reference list to ensure that it is complete and correct. If you have cited papers that have been retracted, please include the rationale for doing so in the manuscript text or remove these references and replace them with relevant current references. Any changes to the reference list should be mentioned in the rebuttal letter that accompanies your revised manuscript. If you need to cite a retracted article, indicate the article’s retracted status in the References list and also include a citation and full reference for the retraction notice.

Reply: The reference list has been reviewed. A correction has been made in the Revised Manuscript with Track Changes, Method section, subheading Statistics, Line 204, from [16] to [23]. This correction did not imply any changes to the reference list. Seven references were added in the first revision to include an introductory paragraph in the introduction regarding the magnitude of diabetes and DR globally and in Sweden. Two references were excluded in the first revision, but these are now included again (Simo 2018 Diabetologia and van der Kreke, PLOS One 2020). Thus, the reference list in this second revision is correct with 34 references. No papers have been retracted.

Comments and replies to Reviewer #1:

1. If the distinction between type 1 and type 2 diabetes did not impact the results, it might not be necessary to separate them strictly. To avoid potential inaccuracies in the manuscript, defining individuals diagnosed with diabetes under the age of 30 as having 'early-onset diabetes' could be a clearer and more precise alternative.

Reply: The suggestion has been adopted. Revised Manuscript with Track Changes, Method section, subheading Study design, line 132-133 and Result section, line 218.

-------------------

2. Apologies for missing this in the initial review, but the title of Figure 1 appears to be just the legend, which seems very unusual. Perhaps the title could be something like 'ETDRS levels at the first visit and the final visit of the patients' to make it clearer.

Reply: A clarification has been made as suggested. Revised Manuscript with Track Changes, Result section, line 236-237.

Comments and replies to Reviewer #2

Purpose ….better change to Background /introduction based on the journal guideline. see the previous comments

Reply: Additional subheadings now clarify the abstract structure. We hope this is sufficient but are not entirely sure because we find abstracts without and with varying subheadings in PLOS One. We have considered the guidelines and the advice from the following sources: https://journals.plos.org/plosone/s/submission-guidelines#loc-abstract

and https://plos.org/resource/how-to-write-a-great-abstract/

-------------------

Change the diabetes patients as “people living with diabetes”

Reply: The term people living with diabetes is a respectful, including and formal term. The entire manuscript has been changed accordingly.

-------------------

The IQR should be stated in range form not the subtraction result, so edit the entire manuscript table

Reply: The IQR has been changed from subtraction to the range form.

-------------------

The study design must be expressed in the study design section

Reply: A clarification has been added. Revised Manuscript with Track Changes, Method section, subheading Study design, line 127.

-------------------

Include the study implication in the discussion section

Reply: Implications have been further clarified. Revised Manuscript with Track Changes, Discussion section, line 348-353.

-------------------

Please use conclusion as separate a title after discussion.

Reply: A subheading has been added as suggested.

---

## [Decision Letter · Decision Letter 2]

18 Feb 2025

Ten years of visual field change in people living with diabetes: A prospective longitudinal study

PONE-D-24-24590R2

Dear Dr. Hellgren,

We’re pleased to inform you that your manuscript has been judged scientifically suitable for publication and will be formally accepted for publication once it meets all outstanding technical requirements.

Kind regards,

Daisuke Nagasato

Academic Editor

PLOS ONE

Additional Editor Comments (optional):

Reviewers' comments:

Reviewer's Responses to Questions

**Comments to the Author**

1. If the authors have adequately addressed your comments raised in a previous round of review and you feel that this manuscript is now acceptable for publication, you may indicate that here to bypass the “Comments to the Author” section, enter your conflict of interest statement in the “Confidential to Editor” section, and submit your "Accept" recommendation.

Reviewer #1: All comments have been addressed

2. Is the manuscript technically sound, and do the data support the conclusions?

Reviewer #1: Yes

3. Has the statistical analysis been performed appropriately and rigorously?

Reviewer #1: Yes

4. Have the authors made all data underlying the findings in their manuscript fully available?

Reviewer #1: Yes

5. Is the manuscript presented in an intelligible fashion and written in standard English?

Reviewer #1: Yes

6. Review Comments to the Author

Reviewer #1: (No Response)

7. PLOS authors have the option to publish the peer review history of their article (what does this mean? ). If published, this will include your full peer review and any attached files.

**Do you want your identity to be public for this peer review?** For information about this choice, including consent withdrawal, please see our Privacy Policy .

Reviewer #1: No

---

## [Editor Report · Acceptance letter]

PONE-D-24-24590R2

PLOS ONE

Dear Dr. Hellgren,

I'm pleased to inform you that your manuscript has been deemed suitable for publication in PLOS ONE. Congratulations! Your manuscript is now being handed over to our production team.

Kind regards,

on behalf of

Dr. Daisuke Nagasato

Academic Editor

PLOS ONE